 

# The p75 neurotrophin receptor in AgRP neurons is necessary for homeostatic feeding and food anticipation

Brandon Podyma[1], Dove-Anna Johnson[1], Laura Sipe[1†], Thomas Parks Remcho[1‡], Katherine Battin[1], Yuxi Liu[2], Sung Ok Yoon[2], Christopher D Deppmann[1]*, Ali Deniz Güler[1]*

[1]Department of Biology, University of Virginia, Charlottesville, United States; [2]Department of Biological Chemistry and Pharmacology, The Ohio State University College of Medicine, Columbus, United States

**Abstract** Networks of neurons control feeding and activity patterns by integrating internal metabolic signals of energy balance with external environmental cues such as time-of-day. Proper circadian alignment of feeding behavior is necessary to prevent metabolic disease, and thus it is imperative that molecular players that maintain neuronal coordination of energy homeostasis are identified. Here, we demonstrate that mice lacking the p75 neurotrophin receptor, p75NTR, decrease their feeding and food anticipatory behavior (FAA) in response to daytime, but not nighttime, restricted feeding. These effects lead to increased weight loss, but do not require p75NTR during development. Instead, p75NTR is required for fasting-induced activation of neurons within the arcuate hypothalamus. Indeed, p75NTR specifically in AgRP neurons is required for FAA in response to daytime restricted feeding. These findings establish p75NTR as a novel regulator gating behavioral response to food scarcity and time-of-day dependence of circadian food anticipation.

*For correspondence:
deppmann@virginia.edu (CDD);
aguler@virginia.edu (ADG)

Present address: †Department of Medicine, University of Tennessee Health Science Center, Memphis, United States; ‡NIH, Laboratory of Malaria and Vector Research, Bethesda, United States

Competing interests: The authors declare that no competing interests exist.

## Introduction

Neuronal circuits originating in the hypothalamus direct behavioral responses to match an organism's perception of scarce or excess energy environments (*Morton et al., 2006*). For example, scarcity challenges organisms to modulate energy homeostasis, conserving the resources they have and seeking out additional food sources around them. A maladaptive relationship between such energy cues and appropriate behavioral responses is a key feature of many eating disorders (*Becker et al., 1999*).

Homeostatic food intake is also impacted by time of day, with laboratory housed mice normally eating most of their food during their nighttime active phase (*Challet, 2019*). Conversely, scarcity due to a limited window of food availability (*e.g.* a prey species emerges to forage for only a few hours) is capable of inducing adaptation of this ostensibly circadian feeding circuit. The context of this timing information is so significant that regularly recurring cycles of food availability can lead organisms to modify their behavior and physiology, changing their locomotor activity, glucocorticoid levels, and body temperature to better match the predicted time of food availability (*Patton and Mistlberger, 2013*). A growing body of evidence suggests that desynchronization of feeding relative to the normal circadian time of eating adversely impacts metabolic health (*Challet, 2019*; *Hatori et al., 2012*; *Pan et al., 2011*; *Sutton et al., 2018*).

While many of the peripheral responses induced by caloric scarcity are known (e.g. elevated glycogenolysis, increased ketone body production), there is a significant gap in our understanding of the neural and molecular mechanisms leading to scarcity-associated behaviors. In response to time

**eLife digest** In many animals, specific types of neurons, such as the hypothalamic hunger neurons, are tasked with regulating food consumption, integrating internal signals of hunger. In general, individuals eat if food becomes available when they are hungry; if food is absent, they will start moving to find new resources. Finally, if food always comes at the same time, animals will increase their activity just before it is delivered.

Neurotrophins are a family of proteins that have many essential roles in the brain. In recent years, they have been shown to interact with the circadian clock, the built-in mechanism that helps animals stay synchronized with the cycle of day and night. A protein known as p75NTR is present in nerve cells, including hypothalamic hunger neurons: there, it helps to relay messages from a neurotrophin which, amongst other roles, controls food intake. However, it was unclear whether p75NTR played a role in regulating feeding behaviors, especially in a circadian manner.

To investigate this question, Podyma et al. genetically engineered a group of mice lacking p75NTR, and a group missing the protein only in their hypothalamic hunger neurons. Both types of mutants had abnormal control of their feeding behavior: compared to normal mice, they fed less (and lost more weight) after they had been deprived of food overnight, or when they faced food shortage over multiple days. In addition, the mutants failed to move more before being fed. However, these feeding patterns were only affected during daytime, while they were preserved at night. These results reveal a new role for p75NTR in hypothalamic hunger neurons.

Dissecting the biological processes that control food intake is key since obesity levels are increasing around the world. In particular, the relationship between food intake and the circadian clock is an important avenue of research as time-restricted diets (where food intake is only allowed during specific periods of the day) are growing in popularity.

restricted feeding (TRF), mice increase their activity in the time window preceding feeding, a phenomenon known as food anticipatory activity (FAA) (*Richter, 1922*). This is the hypothesized output of a putative food entrainable oscillator (FEO), which functions in a comparable manner for entrainment to food as the suprachiasmatic nucleus (SCN) does for entrainment to light (*Stephan, 2002*). Despite the recognition of FAA, the identification and characterization of the anatomic and molecular correlates of the FEO have remained elusive (*Pendergast and Yamazaki, 2018*).

Recently, it has been hypothesized that the FEO may be anatomically dispersed, with at least one component embedded within hypothalamic circuits to alter feeding behavior in response to peripheral energy status (*Pendergast and Yamazaki, 2018*). One of the hypothalamic drivers of feeding that has been implicated in FAA are AgRP neurons of the arcuate hypothalamus. These cells respond to hunger and satiety factors released from peripheral organs and neighboring neurons to drive feeding and associated behaviors (*Aponte et al., 2011*; *Dietrich et al., 2015*; *Krashes et al., 2011*). Strikingly, neonatal ablation of AgRP neurons leads to diminished FAA, and more prominently so during the daytime (*Tan et al., 2014*). The basis for how AgRP neurons alter FAA, and indeed how the FEO can be impacted by time of day, as observed in the AgRP neuron ablated animals (*Tan et al., 2014*), remains unknown.

Herein, we examined the role of the p75 neurotrophin receptor (p75NTR, *Ngfr*) in modulating behavioral response to homeostatic challenge. p75NTR is one of the two receptors for brain derived neurotrophic factor, BDNF, which is known to promote satiety and prevent weight gain through its actions in the hypothalamus (*An et al., 2015*; *Liao et al., 2015*; *Xu et al., 2003*). While genetic deletion of p75NTR does not alter locomotor activity or metabolism under normal conditions (normal chow, ad lib), its expression cycles in a circadian manner in the SCN, with an increase during the rest phase, and it regulates the oscillation of glucose and lipid homeostasis genes in the liver (*Baeza-Raja et al., 2013*). Interestingly, metabolically relevant functions of p75NTR have been unmasked when animals are homeostatically challenged. Loss of p75NTR improves glucose and insulin regulation during glucose or insulin tolerance tests (*Baeza-Raja et al., 2012*). In response to high fat diet this unmasking effect is even more evident, with resistance to diet-induced obesity observed when p75NTR is ablated in adipocytes (*Baeza-Raja et al., 2016*). The involvement of neurotrophic factors in hypothalamic feeding circuits, and the identified roles of p75NTR in circadian and metabolic

regulation, together suggest that p75NTR may be a candidate to influence food entrainable behaviors such as FAA.

To better understand the role of p75NTR in feeding behavior we studied the response of *Ngfr*-KO mice to energy deficiency. We found that p75NTR is required to consume normal post-fasting levels of food. Unexpectedly, we also find that p75NTR is necessary for circadian expression of daytime, but not nighttime, FAA. Furthermore, we show that these effects do not require any developmental role of p75NTR, instead requiring it's function in AgRP neurons. This work documents a novel role of p75NTR in the CNS circuitry controlling feeding behavior, and posits it as a sought after molecular player to elucidate circadian regulation of feeding behavior.

## Results

### Germline loss of p75NTR does not alter baseline feeding, activity or metabolism

To establish whether p75NTR is involved in any gross processes of energy homeostasis, we first examined body weight, food intake, and locomotor activity in ad libitum fed mice harboring germline knockout alleles of p75NTR (*Ngfr*-KO) (*Lee et al., 1992*). *Ngfr*-KO mice have similar body weight to control animals (*Ngfr*-WT) into adulthood, with increased variability as wildtype mice gain weight with age (*Figure 1—figure supplement 1A*; *Tables 1* and *2*), and with no differences in total daily activity (*Figure 1—figure supplement 1B*). Additionally, *Ngfr*-KO mice increase nighttime food intake by 4% but total daily ad libitum food intake is not significantly different compared to littermate controls (*Figure 1—figure supplement 1C*). In line with previous reports, ad libitum fed *Ngfr*-KO mice also have similar serum chemistry markers compared to controls, including insulin, leptin and corticosterone, during either the day or night (*Tables 1* and *2*; *Baeza-Raja et al., 2012*). These data suggest that p75NTR does not influence energy homeostasis under ad libitum conditions.

**Table 1.** Serum chemistries exhibit similar daytime changes in fed and fasted Ngfr-KO *mice.*
Serum was collected at ZT4 (Zeitgeber Time) in 12–16 week old ad libitum fed mice or following a 16 hr overnight fast. Data are presented as mean ± SEM. *p<0.05 compared to WT by Student's t-test. #p<0.05 compared to fed state by Student's t-test, exact p-values can be found in *Table 1—source data 1*. n = 8/group.

|  |  | *Ngfr*-WT | *Ngfr*-KO |
| --- | --- | --- | --- |
| ZT4 |  |  |  |
| Body Weight (g) | Fed | 29.8 ± 0.7 | 26.6 ± 1.2* |
|  | Fasted | 23.8 ± 1.3[#] | 22.0 ± 0.7[#] |
| Glucose (mg/dl) | Fed | 380 ± 17.5 | 385 ± 16.4 |
|  | Fasted | 245 ± 19.1[#] | 307 ± 25.7[#] |
| Insulin (ng/ml) | Fed | 0.23 ± 0.1 | 0.37 ± 0.1 |
|  | Fasted | 0.10 ± 0.1 | 0.10 ± 0.1[#] |
| Ketones (mM) | Fed | 0.0 ± 0.02 | 0.0 ± 0.02 |
|  | Fasted | 2.1 ± 0.36[#] | 2.1 ± 0.10[#] |
| Leptin (ng/ml) | Fed | 2.1 ± 0.5 | 0.9 ± 0.2 |
|  | Fasted | 0.8 ± 0.16[#] | 0.9 ± 0.14 |
| Corticosterone (ng/ml) | Fed | 117 ± 12.4 | 126.7 ± 13.4 |
|  | Fasted | 170 ± 5.3[#] | 180 ± 4.1[#] |

The online version of this article includes the following source data for Table 1:
**Source data 1.** p values by Student's t-test for each pair-wise comparison in *Table 1*.

**Table 2.** Serum chemistries exhibit similar nighttime changes in fed and fasted Ngfr-KO *mice.*
Serum was collected at ZT16 in 12–16 week old ad libitum fed mice or following a 16 hr overday fast.
Data are presented as mean ± SEM. *p<0.05 compared to WT by Student's t-test. #p<0.05 compared
to fed state by Student's t-test, exact p-values can be found in *Table 2—source data 1*. n = 8/group.

| | | Ngfr-WT | Ngfr-KO |
|---|---|---|---|
| ZT16 | | | |
| Body Weight (g) | Fed | 29.4 ± 0.4 | 23.9 ± 0.5* |
| | Fasted | 26.2 ± 0.5# | 23.3 ± 0.7* |
| Glucose (mg/dl) | Fed | 299 ± 13.0 | 312 ± 29.2 |
| | Fasted | 145 ± 25.6# | 160 ± 13.6# |
| Insulin (ng/ml) | Fed | 2.0 ± 0.5 | 0.88 ± 0.2 |
| | Fasted | 0.16 ± 0.1# | 0.03 ± 0.1# |
| Ketones (mM) | Fed | 0.1 ± 0.03 | 0.2 ± 0.11 |
| | Fasted | 1.6 ± 0.10# | 1.4 ± 0.13# |
| Leptin (ng/ml) | Fed | 3.3 ± 0.9 | 1.5 ± 0.2 |
| | Fasted | 0.8 ± 0.15# | 0.8 ± 0.12# |
| Corticosterone (ng/ml) | Fed | 123 ± 8.3 | 146 ± 8.5 |
| | Fasted | 163 ± 8.2# | 176 ± 3.7# |

The online version of this article includes the following source data for Table 2:
Source data 1. 1 p values by Student's t-test for each pair-wise comparison in *Table 2.*

## p75NTR is required for homeostatic feeding behavior in a time dependent manner

Previously reported metabolic phenotypes of *Ngfr*-KO mice have been identified during homeostatic challenges, such as consumption of high fat foods that normally result in diet-induced obesity. For example, loss of p75NTR from white adipose depots results in excess lipolysis and resistance to weight gain on an energy dense diet (*Baeza-Raja et al., 2016*). We thus sought to examine the role of p75NTR under homeostatic challenge, but with energy deficiency rather than energy excess. Weight-matched *Ngfr*-KO mice and wildtype littermate controls were fasted overnight (16 hr) and then refed for 3 hr during the day (*Figure 1A*). During daytime refeeding, we found that *Ngfr*-KO mice consumed ~21% less food than wild-type littermate controls (*Figure 1A*). Since mice eat ~85% of their food during the night (*Figure 1—figure supplement 1C*) and p75NTR expression is under circadian control, we repeated the experiments with daytime food deprivation and nighttime refeeding (*Figure 1B*). Unlike daytime refeeding, *Ngfr*-KO mice consumed similar amounts of food with nighttime refeeding as controls (*Figure 1B*).

As mice normally increase their activity during fasting, purportedly as an effort to forage for food, we additionally monitored locomotor activity during the initial 12 hr of food deprivation to assess the activity response of *Ngfr*-KO mice (*Gelegen et al., 2006*; *Jensen et al., 2013*). In agreement with previous literature we found that wildtype mice trended towards increased nighttime locomotor activity when they were fasting as compared to their activity during ad libitum feeding (*Figure 1C*). However, we observed that *Ngfr*-KO mice decreased their fasted nighttime activity relative to controls (*Figure 1C*). In contrast, both wildtype and *Ngfr*-KO mice decreased their fasted daytime activity (*Figure 1D*). These results suggest that *Ngfr*-KO mice may have defective responses to hunger that extend beyond feeding behavior.

We next sought to determine whether the perturbations observed in feeding and locomotor activity reflect changes of peripheral hunger signals that would lead to alterations of central hunger responses. To this end, we assessed serum hormone and nutrients in both the daytime and nighttime fasted state. We found comparable levels of most of the measured peripheral metabolites between wildtype and *Ngfr*-KO mice (*Tables 1* and *2*), with similar regulation between fed and fasted states. Of note, there is a lack of a drop in leptin levels in overnight fasted *Ngfr*-KO mice, which may contribute towards the observed feeding defect. Together with the observed behavioral

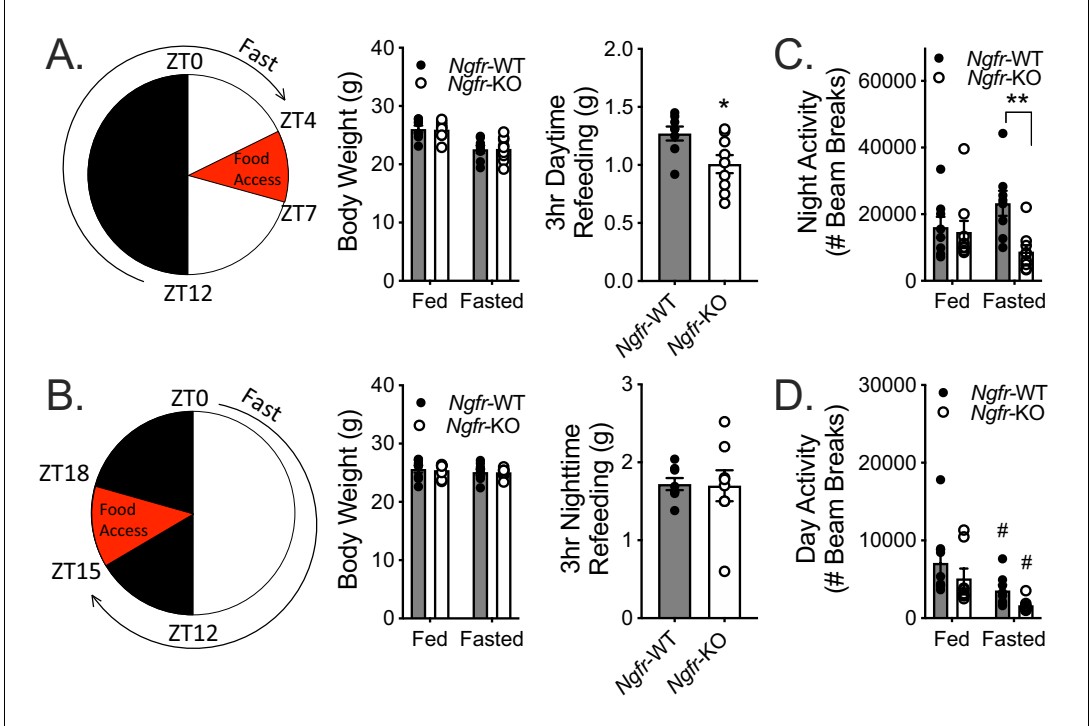

**Figure 1.** Fasting Ngfr-KO mice reveals activity and feeding deficits. (**A**) Schematic of nighttime fasting (left). Mice are fasted just before lights out at ZT12, weighed at ZT0, and fed for 3 hr between ZT4 and ZT7 (right). n = 8 WT, 9 KO. *p=0.0195 by Student's t-test. (**B**) Schematic of daytime fasting (left). Mice are fasted just after lights on at ZT0, weighed at ZT12, and fed for 3 hr between ZT15 and ZT18 (right). n = 8 WT, 9 KO. Not significant by Student's t-test. (**C**) Locomotor activity during the 12 hr night (initial 12 hr of fasting) was measured in ad libitum fed and fasted mice, as described in A. n = 8 WT, 9 KO. **p=0.0045, F(1,15)=5.443 by two-way repeated measures ANOVA with Bonferroni multiple comparisons. (**D**) Locomotor activity during the 12 hr daytime (initial 12 hr of fasting) was measured in ad libitum fed and fasted mice, as described in B. n = 8/group. #p=0.02 versus fed, F(1,14) =16.44 by two-way repeated measures ANOVA with Bonferroni multiple comparisons. All mice are age and weight-matched. Data are presented as mean ± SEM.

The online version of this article includes the following figure supplement(s) for figure 1:

**Figure supplement 1.** Germline loss of p75NTR does not alter body weight, activity, or daily food intake.

**Figure supplement 2.** Refeeding food intake normalized to total body weight.

alterations, these data suggest that p75NTR is necessary for appropriate responses to changes in energy balance, and suggest the feeding and activity responses may be functioning in a time-of-day dependent capacity.

## Daytime food anticipation requires p75NTR

The aforementioned phenotypes of decreased nighttime activity during fasting and daytime refeeding prompted the question: Is p75NTR required for food anticipatory activity (FAA)? FAA is measured as an increase in locomotor activity in the timeframe preceding a scheduled meal, indicating a preparedness to consume food quickly in times of caloric scarcity (*Gallardo et al., 2014*; *Pendergast and Yamazaki, 2018*). To investigate FAA behavior of mice we subjected *Ngfr*-KO mice and littermate controls to the paradigm illustrated in *Figure 1A* for 5 days. Body weight, blood glucose, and ketones were measured at ZT0 (*Figure 2A*). In agreement with our finding that *Ngfr*-KO mice have diminished refeeding following an overnight fast (*Figure 1A*), on each of the five days of daytime TRF, *Ngfr*-KO mice consumed significantly less food and lost more body weight than littermate controls (*Figure 2B*), with no detectable differences in blood glucose or ketone levels (*Figure 2—figure supplement 1*). When we assessed activity, littermate controls showed a robust increase in the proportion of their daytime activity that occurred before feeding (FAA), while *Ngfr*-KO mice showed no such increase in their FAA after five days (*Figure 2C*). Furthermore, this lack of

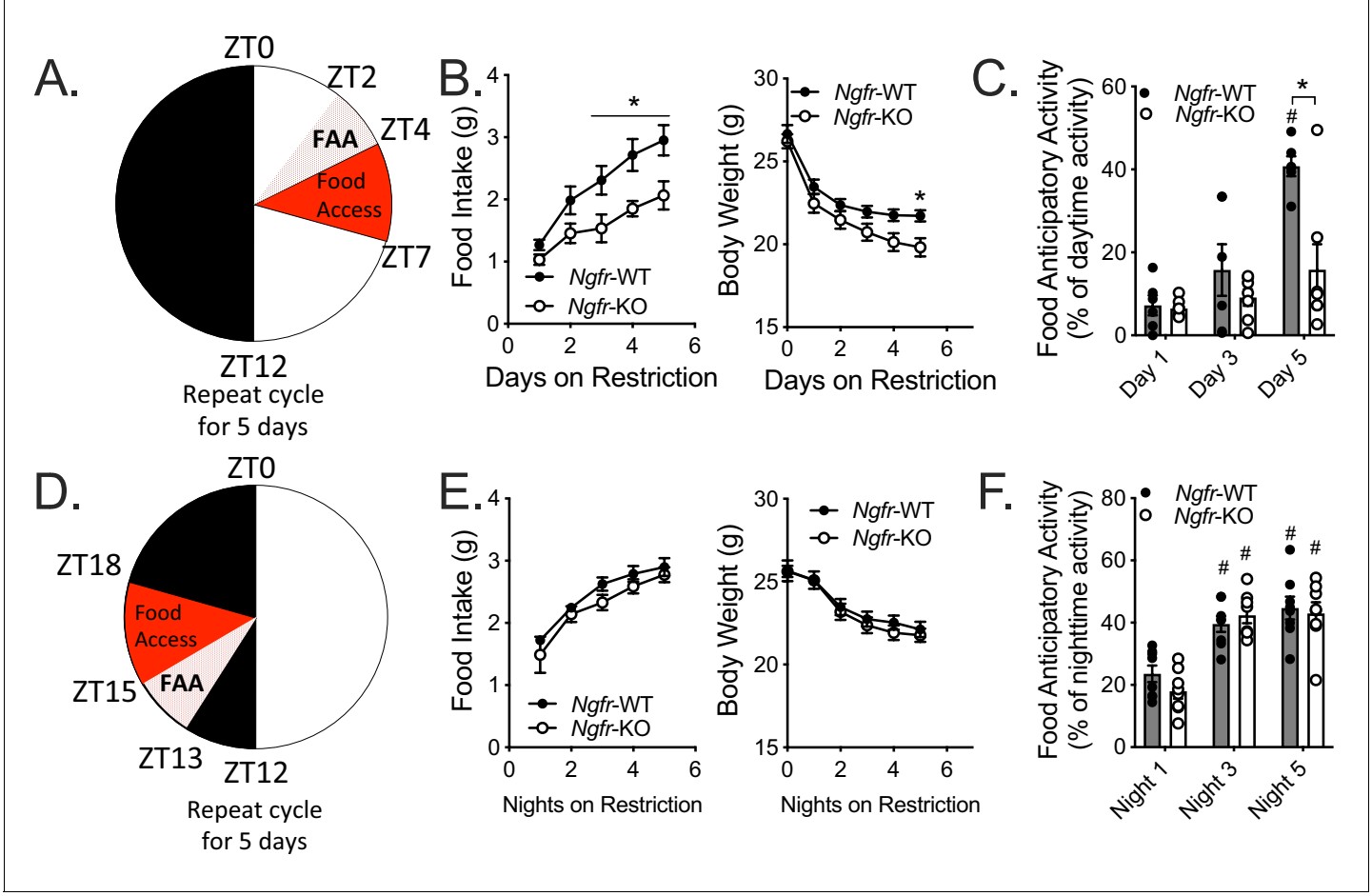

**Figure 2.** Ngfr-KO mice have blunted food anticipatory activity in response to daytime, but not nighttime, restricted feeding. (A) Schematic of daytime restricted feeding paradigm. Mice are fasted just before lights out at ZT12, weighed at ZT0, fed for 3 hr between ZT4 and ZT7, and then fasted again until the next day's feeding period. (B) Food intake (left, $F_{(1,10)}=7.065$) and body weight (right, $F_{(1,10)}=3.64$) on the five consecutive days of restriction. (Left) *p=0.0382 (day 3), 0.0155 (day 4), 0.0122 (day 5), $F_{(1,10)}=7.065$; (Right) *p=0.0358, $F_{(1,10)}=3.64$; by two-way repeated measures ANOVA with Bonferroni multiple comparisons; n = 6/group. (C) Percent of daytime locomotor activity that occurs in the two hour window (ZT2-4) preceding feeding. #p<0.0001, $F_{(2,22)}=21.21$ vs day 1; *p=0.0169, $F_{(1,11)}=7.613$ vs control by two-way repeated measures ANOVA with Bonferroni multiple comparisons; n = 6 WT, 7 KO. (D) Schematic of nighttime restricted feeding paradigm. Mice are fasted just after lights on at ZT0, weighed at ZT12, fed for 3 hr between ZT15 and ZT18, and then fasted again until the next day's feeding period. (E) Food intake (left) and body weight (right) on the five consecutive nights of restriction. n = 8/group. Not significant by two-way repeated measures ANOVA with Bonferroni multiple comparisons. (F) Percent of nighttime locomotor activity that occurs in the two hour window (ZT13-15) preceding feeding. #p<0.0001, $F_{(2,28)}=41.62$ vs night one by two-way repeated measures ANOVA with Bonferroni multiple comparisons. n = 8/group. All mice are age and weight-matched. Data are presented as mean ± SEM. The online version of this article includes the following figure supplement(s) for figure 2:

**Figure supplement 1.** Glucose homeostasis and ketone production are intact in Ngfr-KO mice during daytime restricted feeding; Normalized food intake.

**Figure supplement 2.** Daytime restricted feeding selectively dampens FAA, while nighttime restricted feeding blunts several activity measures.

activity is specific to the FAA time period, as we do not observe any significant changes in dark period activity after 5 days of TRF (*Figure 2—figure supplement 2A*).

As the strength of FAA differs in response to feeding during the day versus the night (*Davidson et al., 2003*; *Tan et al., 2014*), and since p75NTR has been implicated as a clock gene in the control of circadian and metabolic transcript levels (*Baeza-Raja et al., 2013*), we next modified our 5 day TRF paradigm to give access to food from ZT15-18, during the night (*Figure 2D*). In contrast to daytime TRF, we found that *Ngfr*-KO mice exhibited unaltered food intake and body weight on nighttime TRF (*Figure 2E*) with intact FAA (*Figure 2F*), albeit with a reduction of total activity

(*Figure 2—figure supplement 2B*). These data indicate p75NTR plays a role in the expression of FAA in a circadian phase dependent manner.

## The role of p75NTR in adaptation to energy deficit is independent of its role in nervous system development

p75NTR has been implicated in nervous system development, acting as either a death signal or a survival cue depending on the cell type (*Bamji et al., 1998*; *Cheng et al., 2018*). As such, the behavioral/feeding phenotypes observed could be due to developmental miswiring of central circuits necessary for energy detection and appropriate responses. To address this possibility, we generated adult specific p75NTR knockout mice by treating inducible *Ndor1*$^{Tg(UBC-Cre/ERT2)}$*::Ngfr*-floxed mice with tamoxifen in young adulthood (Adult-*Ngfr*-KO) (*Ruzankina et al., 2007*). During ad libitum feeding we observed no significant difference in body weight or food intake between Adult-*Ngfr*-KO and wildtype controls (*Figure 3A*). However, Adult-*Ngfr*-KO mice exhibited a similar defect in daytime refeeding (*Figure 3B*) and FAA during daytime TRF with a significant reduction in food intake (*Figure 3C,D*). Furthermore, a third of the Adult-*Ngfr*-KO mice had to be removed early due to weight loss in excess of 30% of baseline weight, suggesting that the phenotype of Adult-*Ngfr*-KO

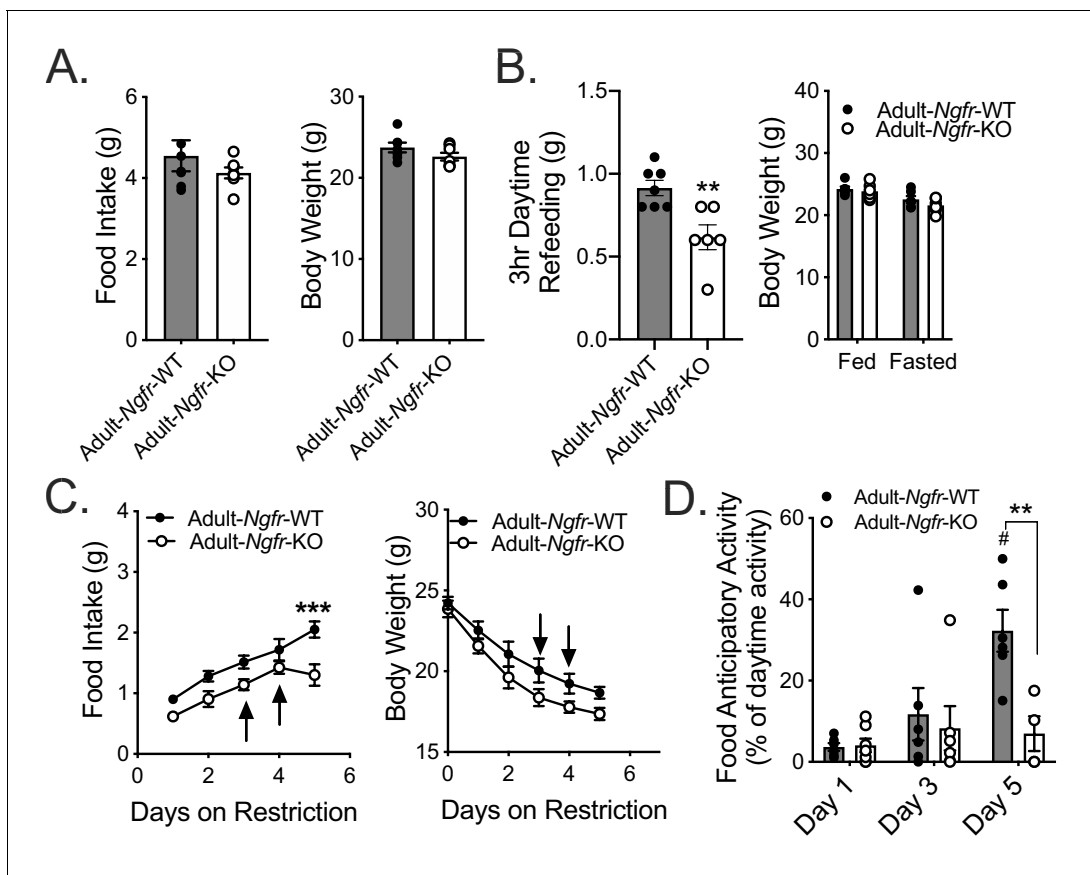

**Figure 3.** Adult loss of p75NTR leads to reduced homeostatic feeding and absent FAA. (**A**) Ad libitum food intake (left) and body weight (right) 4 weeks after tamoxifen injection; n = 7/group. Not significant by Student's t-test. (**B**) Mice are fasted just before lights out at ZT12 and fed for 3 hr between ZT4 and ZT7 (left). They are weighed at ZT0 before and after the fast (right). n = 7 WT, 6 KO. **p=0.005 by Student's t-test. (**C**) Food intake (left) and body weight (right) during daytime restricted feeding paradigm as described in *Figure 2A*; n = 6/group. Arrows indicate mice that were removed from the experiment due to excessive weight loss (>30% of baseline). ***p=0.0002, F(1,10)=10.96 by mixed effects analysis with Bonferroni multiple comparisons. (**D**) Percent of daytime locomotor activity that occurs in the two hour window (ZT2-4) preceding feeding. #p=0.0002, F(2,18)=7.247 vs day 1; **p=0.002, F(1,11)=5.031 vs control by mixed effects analysis with Bonferroni multiple comparisons; n = 6/group. All mice are age and weight-matched in B-D. Data are presented as mean ± SEM.

The online version of this article includes the following figure supplement(s) for figure 3:

**Figure supplement 1.** Adult food intake normalized to total body weight.

mice is more severe than that of the germline knockout (*Figure 3C*, arrows). These results together demonstrate that the *Ngfr*-KO phenotype can be ascribed to an adult function of p75NTR, rather than a developmental one. In addition, this implies that there may be a degree of developmental compensation in germline *Ngfr*-KO mice that rescues homeostatic feeding and food anticipatory behavior.

## p75NTR is expressed in and required for fasting induced activation of the arcuate hypothalamus

Previously identified roles of p75NTR in modulating whole body metabolism have been attributed to its function in white adipocytes and other peripheral tissues (*Baeza-Raja et al., 2016*; *Baeza-Raja et al., 2012*). Since feeding behavior is tightly controlled by central feeding circuits, we next investigated whether p75NTR may also be necessary in neurons of the hypothalamus. Indeed, we detected p75NTR immunofluorescence through an antibody against p75NTR within the arcuate nucleus of the hypothalamus, and quantified co-expression with NPY/AgRP neurons (*Figure 4A*, 69.97 ± 0.022% of NPY+ neurons co-express p75NTR, n = 3). To address whether p75NTR has a functional role in activating these arcuate neurons in response to fasting, we measured expression of the immediate early gene c-Fos in a region of interest masked on *Npy*-GFP expressing neurons of the arcuate of fasted *Ngfr*-KO mice and littermate controls. We found a 33% reduction of *Npy*-GFP

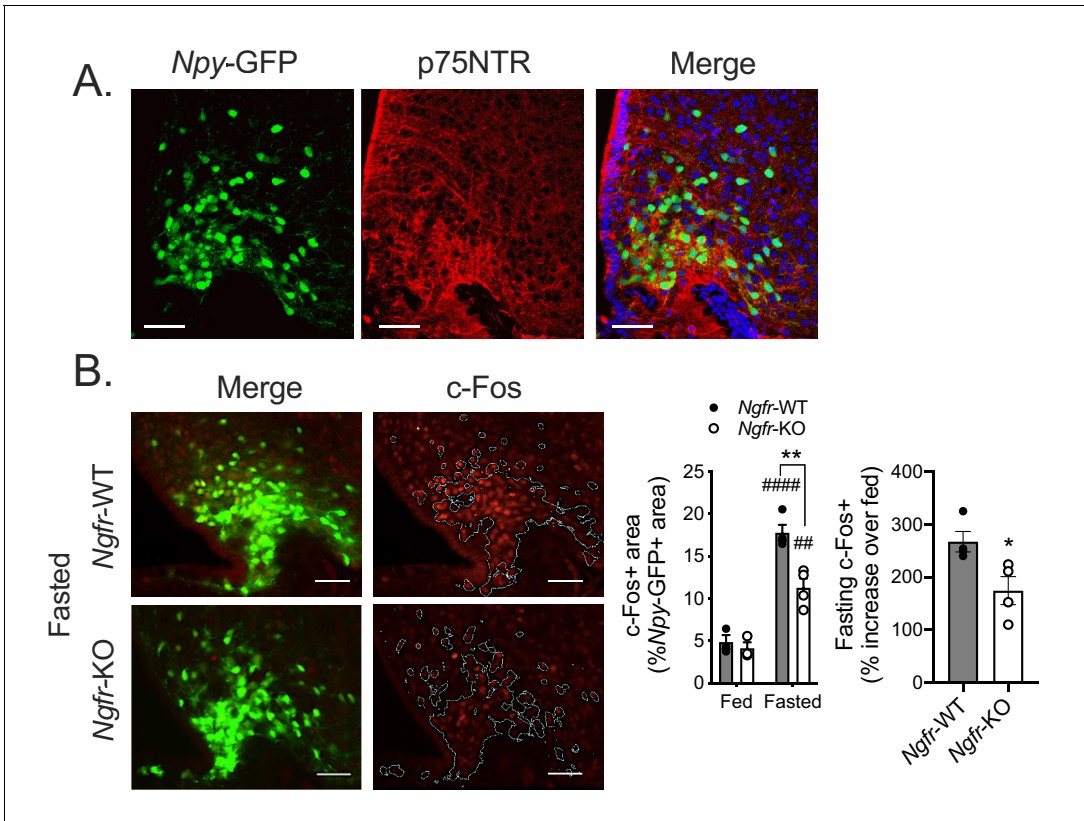

**Figure 4.** p75NTR is expressed in AgRP neurons and is necessary for the arcuate hypothalamus response to fasting. (**A**) Representative images of immunofluorescent staining for p75NTR in an *Npy*-GFP expressing reporter mouse. (**B**) Immunofluorescent staining for c-Fos (left) in 16 hr overnight fasted mice co-expressing an *Npy*-GFP reporter. White lines indicate the approximate boundaries of *Npy*-GFP expression. Quantification (right) of c-Fos expression within *Npy*-GFP expressing areas in fed and fasted mice and the percent increase of c-Fos+/NPY+ areas in each fasted *Ngfr*-WT or *Ngfr*-KO mouse relative to the average of c-Fos+/NPY+ areas of fed *Ngfr*-WT or fed *Ngfr*-KO. Left, ###p<0.0001, ##p=0.0022 vs fed; **p=0.0027, F (1,10)=13.91 vs WT by two-way repeated measures ANOVA with Bonferroni multiple comparisons. Right, *p=0.0302 by Student's t-test. n = 3–4/group. Scale bar = 50 μM. Data are presented as mean ± SEM.

The online version of this article includes the following figure supplement(s) for figure 4:

**Figure supplement 1.** p75NTR is necessary for the arcuate hypothalamus response to fasting in germline and adult knockout mice.

c-Fos activation of fasted *Ngfr*-KO mice, suggesting that p75NTR may be necessary for central detection of energy status (*Figure 4B*). We observed a similar phenotype in arcuate c-Fos induction after fasting in Adult-*Ngfr*-KO mice (*Figure 4—figure supplement 1B*), suggesting a similar defect in these two models. This reduction in c-Fos activation suggests that p75NTR in arcuate neurons themselves may be required for robust neuronal activation during fasting.

## AgRP neurons require p75NTR to promote FAA

As a major driver of homeostatic feeding, NPY/AgRP neurons integrate peripheral and central metabolic information related to energy needs and availability of food (*Betley et al., 2013*; *Su et al., 2017*). In particular, mice with neonatal ablation of AgRP neurons are phenotypically normal during ad libitum feeding, but are unable to increase food intake and express FAA during daytime, but not nighttime, TRF (*Luquet et al., 2005*; *Tan et al., 2014*). This is a remarkably similar phenotype to what we have observed in our *Ngfr*-KO mice on TRF, and led us to suspect that these two mouse models may have defects in the same pathway. Given defective feeding behavior (*Figures 1A* and *2B*), expression in NPY/AgRP neurons (*Figure 4A*), and decreased fasting activation of arcuate neurons (*Figure 4B*), we hypothesized that p75NTR might be necessary for robust function of AgRP neurons. To address this hypothesis, we generated mice with an *Agrp*-specific knockout of p75NTR (AgRP-*Ngfr*-KO). During ad libitum feeding these mice similarly show no defect in food intake or body weight (*Figure 5A*), however exhibit a similar reduction in refeeding food intake following an overnight fast (*Figure 5B*). Furthermore, during daytime TRF, AgRP-*Ngfr*-KO mice behaved similarly to germline *Ngfr*-KO and Adult-*Ngfr*-KO mice, exhibiting no FAA and having reduced food intake (*Figure 5C,D*). However, we do not observe a statistically significant decrease in body weight. Lastly, as AgRP neurons have been shown to be necessary to promote feeding in response to ghrelin, we tested whether p75NTR in AgRP neurons is necessary for ghrelin responsiveness. We found that intracerebroventricular infusion of 1 ug ghrelin at ZT4 in ad libitum fed mice significantly increased food intake and c-Fos expression in both AgRP-*Ngfr*-KO and control mice, suggesting that p75NTR is not required for ghrelin-mediated AgRP neuron activity and feeding behavior (*Figure 5—figure supplement 2*). Taken together, these data suggest that p75NTR is required postnatally and in AgRP neurons for robust behavioral responses to food deficit, while p75NTR in other cell populations may be required for weight loss.

## p75NTR in AgRP neurons is required for induction of phospho-CREB in response to fasting

Roles for p75NTR in the modulation of numerous intracellular signaling pathways have been well documented (*Kraemer et al., 2014*; *Reichardt, 2006*; *Vilar et al., 2009*). One of the primary pathways through which p75NTR has been shown to function is the c-Jun N-terminal kinase (JNK) signaling cascade (*Harrington et al., 2002*; *Yoon et al., 1998*). We first sought to assess whether JNK signaling is perturbed in the arcuate of AgRP-*Ngfr*-KO mice by measuring levels of a JNK signaling target c-Jun through immunofluorescence. We find in the fed state that there is no significant difference in phospho-c-Jun levels between genotypes (*Figure 6A,B*), and that both wildtype and AgRP-*Ngfr*-KO mice show a similar reduction in phospho-c-Jun in response to fasting (*Figure 6A,B*). This suggests that p75NTR is not necessary for JNK activation in AgRP neurons and is in agreement with previous findings that phospho-c-Jun is decreased in AgRP neurons following a fast (*Unger et al., 2010*). We next turned our attention to other signaling pathways that could be altered in p75NTR-deficient AgRP neurons. Phosphorylation of cAMP response element binding protein (CREB) in AgRP neurons is required for adaptive feeding behavior in response to fasting (*Morikawa et al., 2004*; *Yang and McKnight, 2015*). While levels of phospho-CREB are comparable between fed wildtype and AgRP-*Ngfr*-KO mice, we find a significant blunting of fasting-induced phospho-CREB in the arcuate hypothalamus of AgRP-*Ngfr*-KO mice (*Figure 6C,D*), and show this as a specific defect within AgRP neurons (*Figure 6—figure supplement 1*). These data agree with our previous finding showing blunted c-Fos induction in *Ngfr*-KO mice (*Figure 4B*), and suggest that p75NTR is necessary for activation of AgRP neuron CREB signaling in response to fasting.

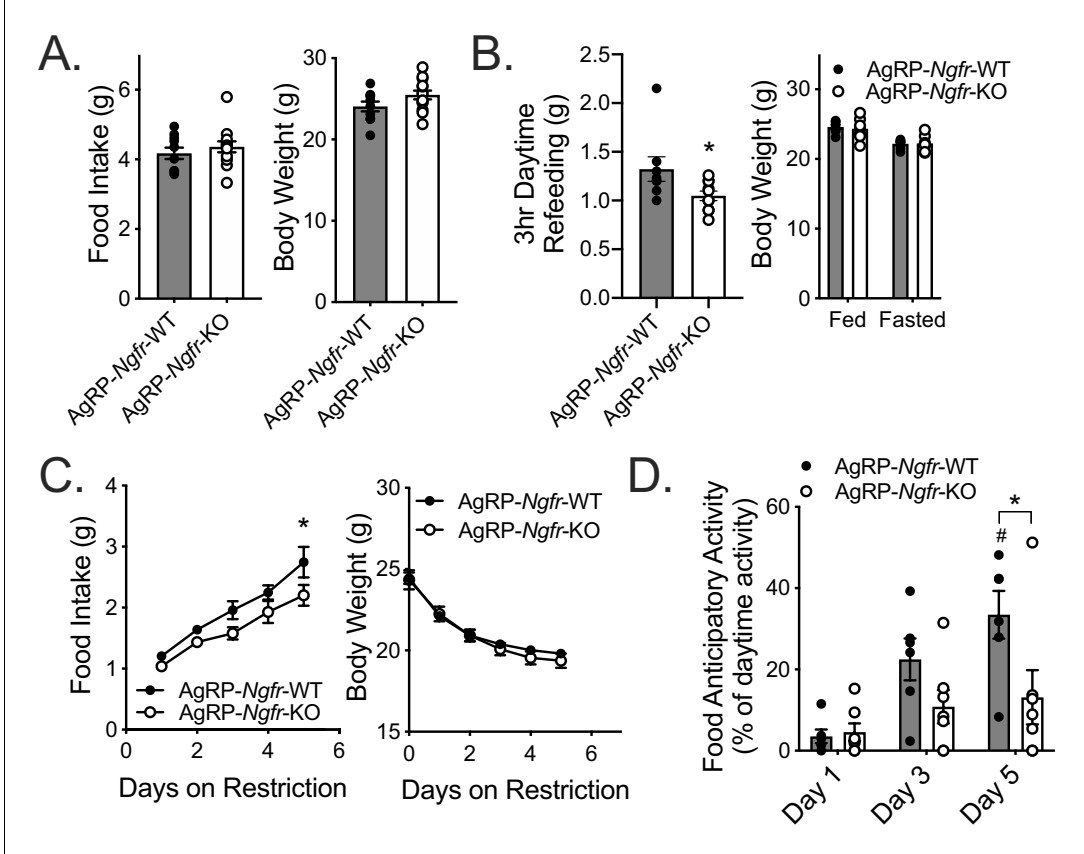

**Figure 5.** AgRP neuron specific loss of p75NTR leads to reduced daytime restricted feeding and FAA. (**A**) Ad libitum food intake (left) and body weight (right); n = 10 WT, 13 KO. Not significant by Student's t-test. (**B**) Mice are fasted just before lights out at ZT12 and fed for 3 hr between ZT4 and ZT7 (left). They are weighed at ZT0 before and after the fast (right). n = 8 WT, 10 KO. *p=0.0416 by Student's t-test. (**C**) Food intake (left) and body weight (right) during daytime restricted feeding paradigm as described in *Figure 2A*. *p=0.0313, F(1,11)=5.024 by two-way repeated measures ANOVA with Bonferroni multiple comparisons; n = 6 WT, 7 KO. (**D**) Percent of daytime locomotor activity that occurs in the two hour window (ZT2-4) preceding feeding. #p=0.001, F(2,22)=8.358 vs day 1; *p=0.013, F(1,11)=7.883 vs control by two-way repeated measures ANOVA with Bonferroni multiple comparisons; n = 6 WT, 7 KO. All mice are age and weight-matched in B-D. Data are presented as mean ± SEM.

The online version of this article includes the following figure supplement(s) for figure 5:

**Figure supplement 1.** AgRP food intake normalized to total body weight.

**Figure supplement 2.** Ghrelin-induced feeding is intact in AgRP-Ngfr-KO mice.

## Discussion

Control of food intake is essential to maintaining metabolic health, and involves complex behaviors such as recognizing meal times and responding to hunger. In response to energy deficit, we present data that p75NTR in the arcuate hypothalamus modulates feeding behavior. This is not due to a developmental effect of p75NTR since adult ablation recapitulates the behaviors observed in the germline knockout. We further demonstrate that loss of p75NTR affects circadian influenced feeding and activity, exerting a predominant effect on daytime feeding behavior. Moreover, we show that p75NTR is expressed in, and modulates the activity of arcuate neurons. Interestingly, this neuronal activation appears to be necessary solely in the context of the homeostatic challenge of hunger, as we observe differences in daytime feeding only in response to an overnight fast. Ablation of p75NTR specifically in AgRP neurons of the arcuate phenocopies the germline knockout, thereby identifying the site of p75NTR's requirement for regulating homeostatic feeding and anticipatory behaviors. Finally, we identify p75NTR in AgRP neurons as necessary for mediating fasting-induced arcuate CREB signaling, and suggest that disruption of this pathway is at least in part responsible for the feeding behavior alterations observed in *Ngfr*-KO mice.

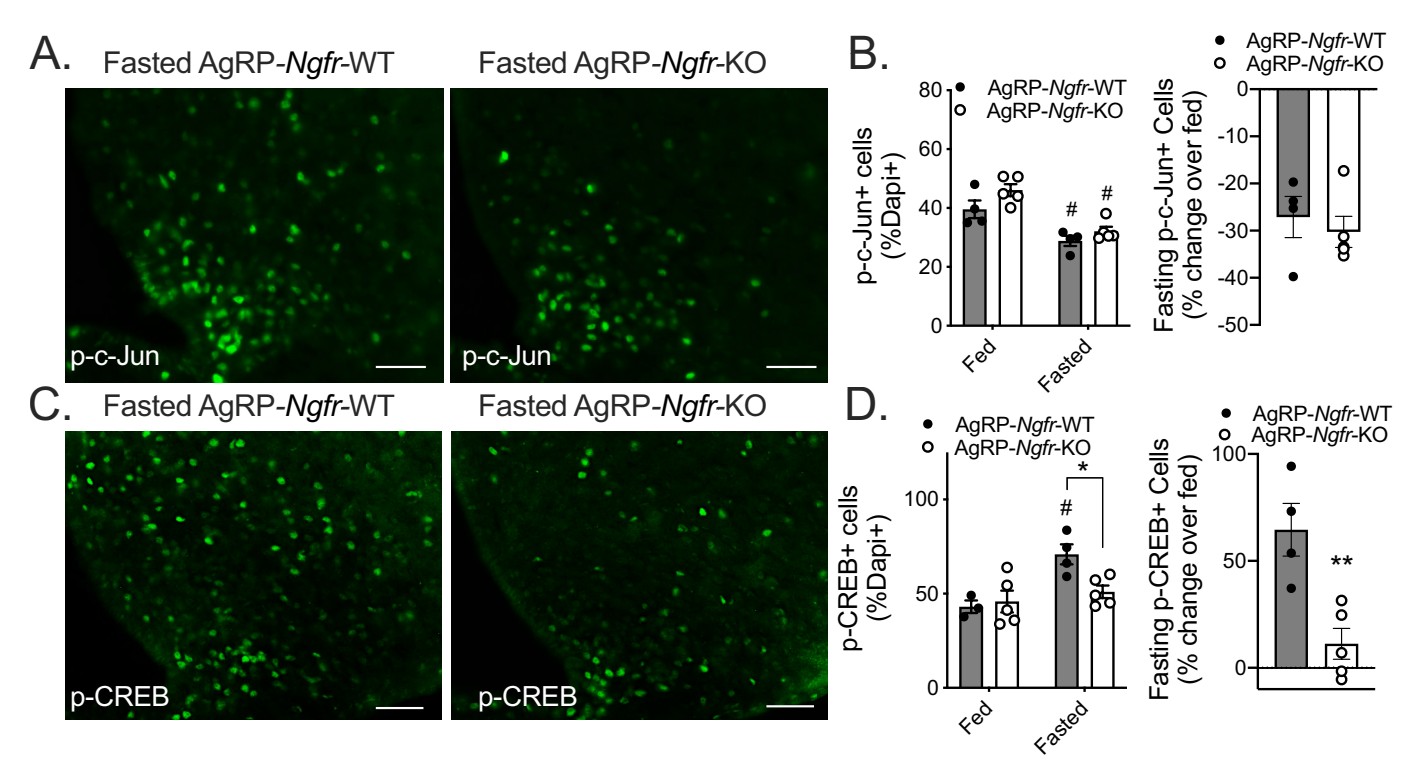

**Figure 6.** AgRP-Ngfr-KO mice have intact fasting JNK signaling, but blunted CREB activation. (**A**) phospho-c-Jun immunofluorescence in fasted AgRP-*Ngfr*-KO mice and littermate controls. Scale bar = 50 µM. (**B**) Quantification of the number of phospho-c-Jun+ cells/dapi+ cells in fed and fasting conditions (left), and the percent increase of phospho-c-Jun+ cells/dapi+ cells in each fasted AgRP-*Ngfr*-WT or AgRP-*Ngfr*-KO mouse relative to the average of phospho-c-Jun+ cells/dapi+ cells in fed AgRP-*Ngfr*-WT or fed AgRP-*Ngfr*-KO (right). n = 4 WT, 5 KO, #p<0.05 compared to fed state. F(1,14)=0.5834 by two-way ANOVA with Bonferroni multiple comparisons in B, left. B, right, not significant by Student's t-test. (**C**) phospho-CREB immunofluorescence in fasted AgRP-*Ngfr*-KO mice and littermate controls. Scale bar = 50 µM. (**D**) Quantification as the number of phospho-CREB+ cells/dapi+ cells in each condition (left), and the percent increase of phospho-CREB+ cells/dapi+ cells in each fasted AgRP-*Ngfr*-WT or AgRP-*Ngfr*-KO mouse relative to the average of phospho-CREB+ cells/dapi+ cells in fed AgRP-*Ngfr*-WT or fed AgRP-*Ngfr*-KO (right). n = 3–4 WT, 5 KO, *p=0.0216 compared to wildtype, #p=0.0174 compared to fed state. F(1,13)=5.266 by two-way ANOVA with Bonferroni multiple comparisons in d, left. **p=0.0057 by Student's t-test in d, right.

The online version of this article includes the following figure supplement(s) for figure 6:

**Figure supplement 1.** AgRP-Ngfr-KO mice have blunted activation of fasting CREB in AgRP neurons.

## How are homeostatic feeding behaviors regulated by circadian rhythmicity?

While hunger has long been recognized as a strong motivator of feeding, it has become clear that circadian inputs may also contribute significantly to driving feeding behavior. In *Ngfr*-KO mice, we show a striking phenotypic difference in the ability to consume equivalent post-fasting amounts of food depending on time of day, with a 21% deficit during the normal rest phase (*Figure 1*). Similarly, the distinction between intact nighttime FAA and lost daytime FAA (*Figure 2*) is intriguing, and suggests that food anticipation may depend on separate mechanisms depending on the phase of TRF. These data suggest that p75NTR may be engaged by hunger cues to override inhibitory signals of feeding that are normally present during the daytime, but which may be absent at night. However, it is unknown how the presence of light or the time-of-day may influence a food entrained clock. These signals could ultimately be derived from the central pacemaker in the SCN, which is known to be more active during the normal rest phase (*Inouye and Kawamura, 1979*), to connect to feeding control centers in the hypothalamus, including the arcuate (*Guzmán-Ruiz et al., 2014*), and function as the master circadian clock in response to changes in light (*Güler et al., 2008*; *Mistlberger, 2011*).

However, it is unresolved how the SCN may interact with a food entrained clock (*Storch and Weitz, 2009*).

AgRP neurons have a diurnal firing pattern (higher during the evening and lower during the morning) which accompanies differential transcriptional profiles in response to refeeding during the day versus at night (*Cedernaes et al., 2019*). Interestingly, among these changes were a significant enrichment in neurotrophin signaling pathway components (e.g. BDNF, Rac1, Ripk2, Frs2, Rap1 etc.) as determined by the KEGG database. p75NTR has been shown to be important in controlling the expression of core clock genes in the SCN and liver (*Baeza-Raja et al., 2013*), and may have a similar role in AgRP neurons. It is possible that some of these transcriptional changes may also be mediated by the modulation of CREB signaling that we observed in AgRP neurons. p75NTR has been suggested to interact with PKA to alter CREB activity, and, interestingly, CREB has been shown to have roles in both the core circadian clock machinery in the SCN, and in the peripheral metabolic alterations associated with time restricted feeding (*Asher and Schibler, 2011*; *Baeza-Raja et al., 2016*; *Ginty et al., 1993*; *Hatori et al., 2012*; *O'Neill et al., 2008*). However, our understanding is limited on how changes in clock genes in metabolically sensitive neurons, such as AgRP neurons, could alter behavioral responses.

## What is the role for p75NTR and other neurotrophins in feeding and circadian biology?

Neurotrophins function broadly in the development and maintenance of nervous system wiring. It could be considered that p75NTR, canonically involved in synaptic plasticity, may influence feeding by altering some broad measure of AgRP neuron remodeling in response to fasting. Indeed, fasting induced activation of AgRP neurons has been shown to require NMDA receptors and spinogenesis (*Liu et al., 2012*). Additionally, CREB signaling, which we show is altered in fasted AgRP-*Ngfr*-KO mice (*Figure 6*), has been shown to be important for long-term changes in neuronal plasticity (*Sakamoto et al., 2011*). Meanwhile, p75NTR can localize to dendritic spines, and loss of p75NTR has been shown to impair NMDA-dependent LTD in the hippocampus (*Woo et al., 2005*). It is intriguing to speculate that the requirement of p75NTR for proper activation of arcuate neurons in response to fasting (*Figures 4* and *6*) is due to a role in dendritic remodeling of AgRP neurons, which will be explored in the future.

It is also possible that neurotrophin family members like p75NTR play non-canonical roles as essential detectors of energy state and in turn regulate feeding behavior. p75NTR, along with TrkB, is one of the two receptors for the neurotrophic factor BDNF, which has been previously implicated in hypothalamic circuits to suppress feeding (*Ozek et al., 2015*). Here we report that p75NTR acts oppositely to BDNF-TrkB signaling to promote feeding. Additionally, mice lacking the ability to form mature BDNF have intact FAA (*Krizo et al., 2018*), suggesting that BDNF signaling, either through TrkB or p75NTR, may not be required for FAA. Interestingly, previous work in the peripheral nervous system has delineated several dichotomous functions for p75NTR and TrkB (*Lu et al., 2005*). The results presented here further support this notion by demonstrating that p75NTR acts in support of feeding, as opposed to TrkB's previously documented suppression of feeding (*Ozek et al., 2015*). It is plausible that p75NTR and TrkB play opposing roles in the coordination of hypothalamic mediated energy coordination. This distinction is made even more notable by recent work demonstrating a time-of-day dependent action of hypothalamic TrkB neurons (*Liao et al., 2019*). Silencing TrkB neurons in the dorsomedial hypothalamus (DMH) during the day, for example, significantly increases food intake, whereas their silencing at night has no effect (*Liao et al., 2019*). While it remains unknown whether p75NTR and TrkB might interact at either a cellular or circuit level in the hypothalamus, we can conclude that neurotrophin receptors have critical functions in time-of-day dependent feeding behaviors.

Understanding the connections between circadian rhythms and metabolism could lead to delineation of how food entrained clocks impact metabolic health. First, deciphering the neuronal regions that mediate diurnal control of feeding and food anticipatory behavior would greatly improve our ability to study this phenomenon. Interactions between the SCN, DMH, and arcuate hypothalamus have long been hypothesized as mediators of circadian control of feeding. Extending our understanding of how neuronal activity in these regions changes normally across the 24 hr light/dark cycle, and how this is altered by changes in feeding patterns, would lend great insight into their circadian regulation. Second, studying the role of important molecular players in circadian regulation of

feeding, such as p75NTR, holds promise to find viable targets for modulating these behaviors. Exploring the role of p75NTR in other hypothalamic neuronal populations will allow further elucidation of the mechanisms of FAA, thereby allowing us to define the neural correlates of how caloric scarcity and time of day work in concert to influence feeding and related behaviors.

# Materials and methods

## Key resources table

| Reagent type (species) or resource | Designation | Source or reference | Identifiers | Additional information |
|---|---|---|---|---|
| Gene *Mus musculus* | *Ngfr* | | NCBI Gene ID: 18053 | |
| Strain, strain background (*Mus musculus*, males) | B6.129S4-*Ngfr$^{tm1Jae}$*/J Also known as p75$^{NGFR}$ | Jackson Labs | RRID: IMSR_JAX:002213 | |
| Strain, strain background (*Mus musculus*, males) | *Ngfr$^{tm1.1Vk}$*/BnapJ Also known as p75$^{NTR-Fx}$flox | *Bogenmann et al., 2011* | RRID: IMSR_JAX:031162 | Gift from Brian Pierchala |
| Strain, strain background (*Mus musculus*, males) | B6.Cg-*Ndor1$^{Tg(UBC-cre/ERT2)1Ejb}$*/1J Also known as UBC-Cre-ERT2 | Jackson Labs | RRID: IMSR_JAX:007001 | |
| Strain, strain background (*Mus musculus*, males) | *Agrp$^{tm1(cre)Lowl}$*/J Also known as Agrp-IRES-cre | Jackson Labs | RRID: IMSR_JAX:012899 | |
| Strain, strain background (*Mus musculus*, males) | B6.FVB-Tg(Npy-hrGFP)1Lowl/J Also known as NPY-GFP | Jackson Labs | RRID: IMSR_JAX:006417 | |
| strain, strain background (*Mus musculus*, males) | B6.Cg-*Gt(ROSA)26Sor$^{tm9(CAG-tdTomato)Hze}$*/J Also known as Ai9 | Jackson Labs | RRID: IMSR_JAX:007909 | |
| Antibody | p75NTR, goat monoclonal | Neuromics | Cat # GT15057 | IF 1:5000 |
| Antibody | c-Fos, rabbit polyclonal | Synaptic Systems | Cat # 226 003 | IF 1:1000 |
| Antibody | phospho-cJun, rabbit monoclonal | Cell Signaling Technology | RRID: AB_2129575 | IF 1:800 |
| Antibody | phospho-CREB | Cell Signaling Technology | RRID: AB_2561044 | IF 1:800 |
| Commercial assay or kit | Insulin elisa | Crystal Chem | RRID: AB_2783626 | |
| Commercial assay or kit | Leptin elisa | Cayman Chemical | Cat # 10007609 | |
| Commercial assay or kit | Corticosterone elisa | Cayman Chemical | Cat # 501320 | |
| Commercial assay or kit | Glucose meter | Bayer | One touch ultra 2 | |
| Commercial assay or kit | Ketone monitor | Abbott | Precision xtra | |
| Peptide, recombinant protein | Ghrelin | Phoenix Pharmaceuticals, inc | Cat # 031–31 | ICV 1 ug |
| Software, algorithm | Prism 8 | Graphpad | RRID: SCR_002798 | |
| Software, algorithm | FIJI | FIJI | RRID: SCR_002285 | |
| Other | DAPI stain | Southern Biotech | Cat # 0100–20 | |
| Other | Opto M4 Activity Monitor | Columbus Instruments | | |

## Mice

All experiments were carried out in compliance with the Association for Assessment of Laboratory Animal Care policies and approved by the University of Virginia Animal Care and Use Committee. Animals were housed on a 12 hr light/dark cycle with food (Teklad Diet 8664) and water ad libitum unless otherwise indicated. *Ngfr*-KO mice were purchased from Jackson Labs (Bar Harbor, Maine) (#002213) (*Lee et al., 1992*), and were maintained on a B6;129 s mixed background and genotyped with primers against intron II in *Ngfr* - Intron II (*Ngfr* -IntII, 5′-CGA TGC TCC TAT GGC TAC TA), Intron III (*Ngfr* -IntIII, 5′-CCT CGC ATT CGG CGT CAG CC), and the pGK-Neo cassette (pGK, 5′-GGG AAC TTC CTG ACT AGG GG). *Ngfr*<sup>fl/fl</sup> mice were acquired as a generous gift from Brian Pierchala (University of Michigan) (*Bogenmann et al., 2011*) and were maintained on a 129/S2/SvPas; C57Bl/6J mixed background and genotyped with a three primer system to detect the wildtype, floxed, and delta alleles (which are generated from unintended germline excision of the loxP sites) using two forward primers (5′-TGC AGA AAT CAT CGA CCC TTC CC), (5′-CCT CCG CCA GCT GTC TGC TTC CT) and a reverse primer (5′-TCC TCA CCC CGT TCT TTC CCC). *Ndor1*<sup>Tg(UBC-Cre/ERT2)</sup> mice (expressing Cre recombinase fused to ERT2 from the ubiquitin C promoter) were purchased from Jackson labs (#008085) (*Ruzankina et al., 2007*). Nuclear translocation of the Cre fusion protein was induced by tamoxifen injections once daily for 5 days in both Adult-*Ngfr*-WT and Adult-*Ngfr*-KO mice (75 mg tamoxifen/kg body weight, Sigma) when mice were 8–9 weeks of age, followed by a 2 week waiting period to ensure excision of floxed alleles (*Heffner, 2011*). *Agrp-IRES-Cre* mice (expressing Cre recombinase from AgRP neurons) were purchased from Jackson labs (#012899) (*Tong et al., 2008*). All Cre recombinase expressing lines were genotyped with primers against the Cre allele (5′-GCA TTA CCG GTC GAT GCA ACG AGT GAT GAG and 5′-GAG TGA ACG AAC CTG GTC GAA ATC AGT GCG) and an internal control sequence (5′-TGG GCT GGG TGT TAG CCT TA and 5′-TTA CGT CCA TCG TGG ACA GC). *Npy*-GFP mice were purchased from Jackson labs (#006417) (*van den Pol et al., 2009*). Ai9 tdTomato mice were purchased from Jackson labs (#007909) (*Madisen et al., 2010*). All experiments were performed on male mice 12–16 weeks old unless otherwise indicated.

## Body weight, Food Intake, and Locomotor Activity

Food intake was performed on individually housed male mice that were acclimated for 7 days, followed by food and body weight measurements weekly (for development curves), twice daily for day/night food intake measures, or daily during restricted feeding experiments. Total and ambulatory activity levels were measured using IR beam interruption (Columbus Instruments).

## Time Restricted Feeding

For scheduled feeding, mice were first acclimated to single housing for 7 days, followed by acclimation to IR beam interruption chambers (Columbus Instruments) for 72 hr. For daytime scheduled feeding, mice were fasted at lights off (ZT12) on day 0. Mice were weighed, and glucose (one touch ultra 2, Bayer, Leverkusen, Germany) and β-ketone (Precision xtra, Abbott, Chicago, Illinois) measures were taken 12 hr later at lights on (ZT0). Mice were then refed 4 hr later at ZT4, with food removed 3 hr later at ZT7. Mice were fed between ZT4-7 on each of the next four days. For nighttime scheduled feeding, mice were fasted at lights on (ZT0) on day 0. Mice were weighed 12 hr later prior to lights off (ZT11-12). Mice were then refed 4 hr later at ZT15, with food removed 3 hr later at ZT18. Mice were fed between ZT15-18 on each of the next four days. All groups were age and weight-matched. In accordance with University of Virginia Animal Care and Use Committee guidelines, any mouse that lost 30% or more of their body weight was removed from the experiment. This is indicated with an arrow in the figure.

## Serum chemistry measurements

Glucose levels were measured using the one touch ultra two glucometer (Bayer). B-ketone levels were measured using the precision xtra meter (Abbott). Insulin was measured using the ultra sensitive mouse insulin ELISA kit according to manufacturer's instructions (Crystal Chem, Elk Grove Village, Illinois). Leptin was measured using the mouse/rat leptin EIA kit according to manufacturer's instructions (Cayman Chemical, Ann Arbor, Michigan). Corticosterone was measured using the corticosterone ELISA kit according to manufacturer's instructions (Cayman Chemical).

## Immunofluorescence

Mice were transcardially perfused by first anesthetizing with ketamine/xylazine, then perfusing with ice cold 1x PBS, followed by ice cold 4% paraformaldehyde (PFA). Brains were removed and placed into 4% PFA overnight, before transitioning to 30% sucrose to dehydrate the brains for 48–72 hr. Brains were then frozen and sliced into 30 micron free-floating sections on a cryostat into 1x PBS with 0.002% sodium azide. Antibody staining was performed as follows: sections were washed in 1x PBS with 0.5% Triton, blocked in 1x PBS with 0.5% Triton and 5% donkey serum, incubated in block with primary antibody overnight at 4C (p75NTR, goat, 1:5000, Neuromics cat# GT15057, Edina, Minnesota; c-Fos, 1:1000, rabbit, Synaptic Systems cat# 226 003, Goettingen, Germany [*Grippo et al., 2017*]; phospho-c-Jun, 1:800, rabbit, Cell Signaling Technologies cat# 3270; phospho-CREB, 1:800, rabbit, Cell Signaling Technologies cat# 9198), again washed, incubated in block with secondary antibody for 2 hr at room temperature (AF 568 donkey anti-goat, 1:500 (for p75NTR); AF 488 or 568 donkey anti-rabbit, 1:500 (for c-Fos), AF 488 donkey anti-rabbit, 1:500 (for phospho-c-Jun and phospho-CREB)), again washed, and mounted with DAPI-fluoromount G (Southern Biotech). Sections were imaged and quantified by a researcher blinded to the genotype and treatment, and unblinded once all analysis was completed. For sections with *Npy*-GFP or *Agrp*-tdTomato reporter co-expression, FIJI software was used to first set a threshold that accounted for reporter expression, and then create a mask of the thresholded region that was overlaid onto c-Fos expression. Surface area of c-Fos expression within the designated region above a threshold determined by a blinded observer was then measured, and normalized to the masked area of the reporter expressing region of interest (*Rieux et al., 2002*).

## Stereotactic surgery

Briefly, animals were anesthetized with isoflurane (induction 5%, maintenance 2–2.5%; Isothesia) and placed in a stereotaxic apparatus (Kopf). A heating pad was used for the duration of the surgery to maintain body temperature and ocular lubricant was applied to the eyes to prevent desiccation. AgRP-*Ngfr*-KO and littermate control mice were stereotactically implanted with a 6 mm cannula targeted to the mediobasal hypothalamus (Plastics one, Roanoke, VA). Stereotaxic coordinates relative to Bregma: 0 mm lateral, 1.46 mm posterior and −5.8 mm below the surface of the skull. Cannulas were maintained in place by dental cement anchored to one stainless steel screw fixed to the skull. A dummy cannula was inserted to prevent clogging of the cannula. After the surgery, the animals were housed individually for 1 week. All surgical procedures were performed in sterile conditions and in accordance with University of Virginia IACUC guidelines.

## Ghrelin administration

Ad libitum fed, cannulated, AgRP-*Ngfr*-KO and littermate control mice were centrally infused with vehicle or ghrelin (0.5 ug/ul) (Phoenix Pharmaceuticals, cat# 031–31) at a rate of 1 μl/min for a final volume of 2 μl. Infusions occurred from ZT3-5, and food intake was measured hourly for three hours. At least 1 week after initial infusions of vehicle and ghrelin, mice were randomly selected to recieve either vehicle or ghrelin at the same concentration and volume, and were intracardially perfused 90 min after infusion for verification of implant target site and immunofluorescence, as described above.

## Statistical analysis

Data are presented as mean ± SEM. Statistical analysis was carried out using Graphpad Prism version 8.0. Outliers were detected using the ROUT method, Q = 1%. Student's t-test was used to compare single means between genotypes, and 2-way Anova was used to compare genotype by time interactions. Differences were considered significant if $p < 0.05$.

## Acknowledgements

We would like to thank members of the Deppmann and Güler labs for helpful feedback and technical assistance during all stages of this work, as well as Ignacio Provencio for helpful comments editing this manuscript.

## Additional information

### Funding

| Funder | Grant reference number | Author |
|---|---|---|
| Hartwell Foundation | | Christopher D Deppmann |
| National Institutes of Health | T32-GM7267-39 | Brandon Podyma |
| National Institutes of Health | T32-GM7055-45 | Brandon Podyma |
| National Institutes of Health | R01-GM121937 | Ali Deniz Güler |
| National Institutes of Health | RO1-AG055059 | Sung Ok Yoon |

The funders had no role in study design, data collection and interpretation, or the decision to submit the work for publication.

### Author contributions

Brandon Podyma, Conceptualization, Formal analysis, Investigation, Visualization, Methodology; Dove-Anna Johnson, Thomas Parks Remcho, Katherine Battin, Yuxi Liu, Investigation; Laura Sipe, Conceptualization; Sung Ok Yoon, Christopher D Deppmann, Ali Deniz Güler, Supervision, Funding acquisition, Project administration

### Author ORCIDs

Brandon Podyma (iD) https://orcid.org/0000-0002-0849-8726
Christopher D Deppmann (iD) http://orcid.org/0000-0002-6591-1767
Ali Deniz Güler (iD) https://orcid.org/0000-0001-8218-850X

### Ethics

Animal experimentation: This study was performed in strict accordance with the recommendations in the Guide for the Care and Use of Laboratory Animals of the National Institutes of Health. All of the animals were handled according to approved institutional animal care and use committee (IACUC) protocols (#3795, 3975, 4183, 4191, 4200) of the University of Virginia.

### Decision letter and Author response

Decision letter https://doi.org/10.7554/eLife.52623.sa1
Author response https://doi.org/10.7554/eLife.52623.sa2

## Additional files

### Supplementary files

• Transparent reporting form

### Data availability

All data generated or analyzed during this study are included in the manuscript and supporting files.

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
