## [Decision Letter]

**Acceptance summary:**

Anticipation to feeding is a response during restricted access to food, and is thought to be dependent upon circuits in the arcuate nucleus of the hypothalamus. This study identifies the p75 neurotrophin receptor (ngfr) as a regulatory component in circadian rhythms that influences daytime food anticipatory behavior. These results are novel and unexpected. The revised manuscript demonstrates expression of p75 in the arcuate neurons to control the timing of food intake. The downstream events are questionable, as the regulation of p-CREB is not normally associated directly with p75 signaling.

**Decision letter after peer review:**

Thank you for submitting your article "The p75 neurotrophin receptor in AgRP neurons is necessary for homeostatic feeding and food anticipation" for consideration by *eLife*. Your article has been reviewed by three peer reviewers, and the evaluation has been overseen by a Reviewing Editor and Jonathan Cooper as the Senior Editor. The reviewers have opted to remain anonymous.

The reviewers have discussed the reviews with one another and the Reviewing Editor has drafted this decision to help you prepare a revised submission.

Summary:

Food anticipatory activity (FAA) is an increase in an animal's overall activity in response to temporally restricted access to food. Although FAA is a widely appreciated phenomenon, its neurobiological basis is unknown. Here the authors provide genetic evidence that the p75 neurotrophin receptor (p75NTR) contributes to FAA through signaling in AgRP neurons, and acts in a circadian manner. Moreover, conditional deletion of p75NTR in mature mice caused significant deficits, suggesting a role that is not dependent on developmental alteration of brain structure and function. While these studies address an understudied aspect of feeding regulation and the data are intriguing, weaknesses in the rigor of the experimental design and data interpretation make it hard to evaluate the authors' claims.

Essential revisions:

It is important to quantify colocalization of p75NTR immunoreactivity with NPY-GFP and to specifically examine c-fos induction and CREB activation in AgRP neurons. The arcuate nucleus has many subsets of neurons in addition to AgRP neurons so that c-fos induction and CREB activation could be in non-AgRP neurons. Thus, the data do not demonstrate that fasting activates CREB and neurons through p75NTR-mediated signaling. The authors can use in situ hybridization such as RNAScope to mark AgRP neurons. In addition, how do the authors know the boundary of the arcuate nucleus in Figure 6 images? The two images in Figure 6C are not comparable.

The authors state that they found, "no alterations of peripheral metabolites between wildtype and p75KO mice," yet they only assayed a small number of molecules. This statement should be revised to more accurately reflect the limited assessment of metabolic alterations, or include a more extensive metabolomic analysis. The failure to detect a significant difference in some serum analytes shown in Tables 1-2 likely stems from an underpowered study (n=3 vs. n=8-9 for measurements of food intake in the same groups). While absolute levels of leptin may not differ in the fasted state, it is notable that the decrease in leptin in response to fasting in p75KO mice is severely blunted. The fall in leptin is important to promote the drive to re-feed, so the absence of this adaptation could contribute to the blunted response of KOs to time-restricted feeding.

The evaluation of the contribution of p75 signaling in AgRP neurons is not sufficiently rigorous. The specificity of the effect for AgRP neurons on p-CREB immunoreactivity was not presented. It is critical to understand how AgRP neuronal activity/function is altered because these neurons are known to promote fasting-induced locomotor activity. Examining whether ghrelin can stimulate c-fos and feeding behavior in AgRPp75KOs would help to parse whether p75 contributes to "hunger recognition" pathways or acts independently to promote FAA/responses to hunger (as discussed in paragraph two of subsection “p75NTR is required for homeostatic feeding behavior 159 in a time dependent manner”).

---

## [Author Response]

Essential revisions:It is important to quantify colocalization of p75NTR immunoreactivity with NPY-GFP and to specifically examine c-fos induction and CREB activation in AgRP neurons. The arcuate nucleus has many subsets of neurons in addition to AgRP neurons so that c-fos induction and CREB activation could be in non-AgRP neurons. Thus, the data do not demonstrate that fasting activates CREB and neurons through p75NTR-mediated signaling. The authors can use in situ hybridization such as RNAScope to mark AgRP neurons. In addition, how do the authors know the boundary of the arcuate nucleus in Figure 6 images? The two images in Figure 6C are not comparable.

We have added data to the text in subsection “p75NTR is expressed in and required for fasting induced activation of the arcuate hypothalamus” suggesting that approximately 70% of AgRP neurons are immunoreactive for p75NTR. In Figure 4B we have also included a more detailed analysis of c-Fos changes specifically in NPY-GFP expressing neurons of fed and fasted *Ngfr-KO* and control mice, showing a similar blunting of c-Fos activity. In Figure 6—figure supplement 1, we have also included a more detailed analysis of pCREB signaling in *AgRP-Ngfr-KO* and control mice by quantifying pCREB expression within AgRP neurons using an AgRPcre-tdTomato reporter. In Figure 6 we had estimated the boundaries of the arcuate nucleus based on morphology of dapi stained nuclei, but have removed those boundaries for their lack of precision. We have replaced the images in Figure 6C with images that we believe are more comparable.

The authors state that they found, "no alterations of peripheral metabolites between wildtype and p75KO mice," yet they only assayed a small number of molecules. This statement should be revised to more accurately reflect the limited assessment of metabolic alterations, or include a more extensive metabolomic analysis. The failure to detect a significant difference in some serum analytes shown in Tables 1-2 likely stems from an underpowered study (n=3 vs. n=8-9 for measurements of food intake in the same groups). While absolute levels of leptin may not differ in the fasted state, it is notable that the decrease in leptin in response to fasting in p75KO mice is severely blunted. The fall in leptin is important to promote the drive to re-feed, so the absence of this adaptation could contribute to the blunted response of KOs to time-restricted feeding.

We thank the reviewers for their insight on this point, and now present a sample size of 8 mice per group for the serum measurements presented in Tables 1 and 2, and have also changed our language in paragraph three of subsection “p75NTR is required for homeostatic feeding behavior in a time dependent manner” and subsection “p75NTR is expressed in and required for fasting induced activation of the arcuate hypothalamus” to better reflect the limited nature of molecules assayed. While we still find that the absolute levels of leptin in the fed or fasted states do not differ significantly between genotypes, as the reviewer mentions we do find a lack of a statistically significant drop between fed and fasted state leptin levels in the *Ngfr-KO* mice. This might be partially explained by the lower body weights of the germline knockouts at this age (16 weeks), and may contribute towards the feeding defect of these animals. We have updated the text to reflect this point.

The evaluation of the contribution of p75 signaling in AgRP neurons is not sufficiently rigorous. The specificity of the effect for AgRP neurons on p-CREB immunoreactivity was not presented. It is critical to understand how AgRP neuronal activity/function is altered because these neurons are known to promote fasting-induced locomotor activity. Examining whether ghrelin can stimulate c-fos and feeding behavior in AgRPp75KOs would help to parse whether p75 contributes to "hunger recognition" pathways or acts independently to promote FAA/responses to hunger (as discussed in paragraph two of subsection “p75NTR is required for homeostatic feeding behavior 159 in a time dependent manner”).

As outlined above, we have added data to Figure 6—figure supplement 1, detailing AgRP neuron pCREB activity of fed and fasted *AgRP-Ngfr-KO* and control mice. Further, we have also added data in Figure 5—figure supplement 1, showing that icv delivery of 1ug ghrelin into ad libitum fed *AgRPNgfr- KO* mice and controls is capable of promoting food intake and arcuate c-Fos activity. We have also modified our language to remove reference to “hunger recognition”.